# Possibilities of Implementing Hospital-Based Health Technology Assessment (HB-HTA) at the Level of Voivodeship Offices in Poland

**DOI:** 10.3390/ijerph191811235

**Published:** 2022-09-07

**Authors:** Maciej Furman, Małgorzata Gałązka-Sobotka, Damian Marciniak, Iwona Kowalska-Bobko

**Affiliations:** 1Department of Health Policy and Management, Faculty of Health Sciences, Institute of Public Health, Jagiellonian University Medical College, 31-066 Cracow, Poland; 2Institute of Healthcare Management, Lazarski University, 02-662 Warsaw, Poland

**Keywords:** medical technology, HB-HTA, Voivodeship Office, semi-structured interviews

## Abstract

The Health Technology Assessment is based on the evaluation of the characteristics and effects of health technologies to properly spend resources in healthcare. For the needs of hospitals, a special HTA department, Hospital-Based Health Technology Assessment (HB-HTA), has been established. The objective of the article is to assess the possibility of implementing a functional model with the coordinating role of Health Departments of the Voivodeship Offices with the support of the National Health Fund and the HTA Agency in Poland. Ten semi-structured interviews were conducted with representatives from eight Voivodeship Offices. The interviews consisted of nine questions related to the possibility of introducing a functional model with the participation of the Voivodeship Office. The material was divided into seven codes relating to the questions included in the topic guide. From the perspective of Voivodeship Offices, HB-HTA could contribute to the improvement of the methodology used in the Evaluation Instrument of Investment Motions in Health. The lack of personnel in the Voivodeship Offices was identified as one of the greatest barriers to the implementation of HB-HTA. These public administration units should not be involved in the hospital health technology assessment process.

## 1. Introduction

The rapid development of new medicines, devices, procedures, and care pathways means that the scope of treatment options persist to grow faster than the resources available to many patients and healthcare systems, particularly as the impacts of the global financial crisis are felt. Identifying treatment options that offer value and value for money is therefore becoming increasingly relevant [1].

Health Technology Assessment (HTA) is a multidisciplinary process that uses systematic and transparent methods to consider the best available evidence to determine the value of a health technology at different points in its lifecycle. It consists in the evaluation of the features and effects of a health technology compared with existing alternatives, to inform decision making in order to promote an equitable, efficient, and high-quality health system [2]. To increase the use of evidence in public health decision-making, health technology assessment has been proposed as a promising strategy. HTAs use a multidisciplinary approach to synthesize available evidence about the consequences of an intervention to inform policy and practice. However, HTA processes have not been widely institutionalized for public health [3].

Activities in the field of health technology assessment are aimed at indicating rational premises related to the decision to finance drug and non-drug technologies and treatment methods with the participation of public funds. With the growing use and importance of HTA in decision making during recent years, health technology assessors, decision makers, and stakeholders are confronted with methodological challenges due to specific characteristics of health technologies (e.g., pharmaceuticals, diagnostic tests, screening programs), their developmental environment, and their regulation process [4]. Several countries have therefore developed policy frameworks to allow access to the technology on the condition that additional evidence is generated through “access with evidence generation” mechanisms [5].To meet all the requirements related to the HTA process, which is based on evidence-based medicine (EBM), it was necessary to introduce a separate healthcare institution in Poland, which could deal with this area in a professional and transparent manner, in consonance with domestic and foreign standards [6]. 

Therefore, in 2005, based on the Executive regulation of the Minister of Health, the Agency for Health Technology Assessment was established, which in 2009 obtained a state organizational unit with legal personality functions. According to the Act of 22 July 2014 amending the Act on healthcare services financed from public funds of 1 January 2015, the Agency received additional tasks related to the tariffication of healthcare services. In this way, the Agency for Health Technology Assessment and Tariff System (pol. Agencja Oceny Technologii Medycznych i Taryfikacji, AOTMiT) was established [7]. The scope of analyses carried out by AOTMiT is expanding every year, as evidenced by the increase in the number of employees and orders that are imposed on it in connection with the implementation of new solutions. They are aimed at creating a health policy based on data and reliable scientific information (Figure 1).

### 1.1. The Role of the Region in Healthcare in Poland

The healthcare system in Poland is based on common health insurance. Funds are distributed to healthcare organizations by the National Health Fund, which is a third-party payer contracting providers for publicly funded health services. The prices of health services are homogenous at the country level, and in terms of in-patient services, they are mostly DRG (Diagnostic-Related Group)-based. Most of the providers in the hospital sector are public hospitals, which can be owned by regional authorities, the Ministry of Health, or universities. All citizens are entitled to receive publicly funded health services, provided that those services are included in the “basket of health services”. One of the conditions for including the health service in “the basket” is the positive approval of the health technology by the HTA Agency [8].

The healthcare management system in Poland is divided into many levels. The responsibility in this area is taken on by representatives of central authorities and local governments. At the Voivodeship level, there is both the representation of the first in the form of Voivodes and the second in the form of Voivodeship Marshals together with Executive Council (pol. Zarząd Województwa) and the residence assembly (pol. Sejmik Województwa) [9].

In the context of healthcare regulations, the Voivode performs the tasks of the government administration within the “Health” section. It includes, in particular, health protection and its organization, supervision over medicinal products and medical devices, organization of the State Medical Rescue, organization, and implementation of the supervisory function over the performance of medical professions, registration of medical entities, supervision over sanitary inspection and spa treatment, as well as determination of the Maps of Health Needs or establishing a regional transformation plan [10]. The last two tasks have been delegated to this authority recently, which allows to present the thesis that voivodes are to act as coordinators of the healthcare system at the Voivodeship level [11].

One of the specified areas of the voivode’s activity is also the assessment of the legitimacy of investments carried out in hospitals. This task is carried out by the activities of the employees of the Health Department by means of a special Evaluation Instrument of Investment Motions in Healthcare (pol. Instrument Oceny Wniosków Inwestycyjnych w Sektorze Zdrowia, IOWISZ). The applicant responds to a few issues related to the assessed investment, including its type, place of implementation, values, and technical and economic indicators. In its assumption, the completion of the application is to prevent the chaotic and short-sighted development of the medical services market while increasing the efficiency of spending public funds [12].

The IOWISZ application is submitted to the voivode in the case of: creation of a new medical entity, new organizational units or cells of the healthcare institution of the healthcare entity, and other investment, the estimated value of which exceeds 2 million at the date of submission of the application. A positive opinion on the advisability of the investment is valid for 3 years from the date of its issuance. An opinion application, the estimated value of which exceeds 50 million, is submitted to the Ministry of Health [13].

At the local government level, the Voivodeship is represented by the Marshal of the Voivodeship, and the authorities of this level of authority are the Executive Council and the residents assembly. Its own tasks in the field of ensuring equal access to health care services provided by the Voivodeship self-government include, in particular, the development, implementation, and assessment of the effects of health policy programs, resulting from the identified health needs and health condition of the Voivodeship inhabitants after consultation with relevant competent communes and counties, inspiring and promoting solutions to increase efficiency, including restructuring in healthcare. The self-government of the Voivodeship is also the owner of medical entities, through which it carries out the task of ensuring proper access to health care services for residents [14].

### 1.2. HB-HTA

Due to the development of the HTA methodology, it currently applies not only to drug reimbursement decisions, but also, among others, to technologies implemented in healthcare facilities. For this reason, a separate part of HTA was created, i.e., Hospital-Based Health Technology Assessment. One of the milestones in the dissemination of HB-HTA around Europe and in the world was the implementation of the AdHopHTA project (Adopting Hospital-Based Health Technology Assessment), the aim of which was to disseminate HTA in hospital settings. To facilitate the implementation of HB-HTA, the project provided pragmatic knowledge and implementation tools: manual, toolkit, and databases. On the other hand, the impetus for the development of HB-HTA in Poland is the project entitled “Implementation of Hospital Assessment of Innovative Medical Technologies”, financed under the grant from the National Center for Research and Development (project number: 1/395107/18/NCBR/2018), led by the consortium of the National Health Fund, the National Institute of Cardiology, and the Lazarski University. The main assumptions of the project work are the implementation of the HB-HTA methodology, along with the creation of pilot HB-HTA units and HB-HTA networks supporting them (which would consist of other healthcare institutions), as well as increasing the possibilities of managing the healthcare system in Poland [15].

In connection with the design work, six functional models of HB-HTA implementation in Poland were proposed. The functional model is a detailed description of the role of individual health care institutions within HB-HTA. The solutions for the implementation of the Hospital-Based Health Technology Assessment have been specified: with the coordinating role of: (1) hospital; (2) Regional Center for Investment Assessment (pol. Regionalne Centrum Oceny Inwestycji, RCOI); (3) AOTMiT, (4) National Health Fund (NHF), (5) external unit, and (6) RCOI at the level of Voivodeship Offices with the support of the National Health Fund and AOTMiT (mixed model). As a result of expert meetings with hospital directors, heads of Health Departments in Voivodeship Offices, and employees of AOTMiT, the National Health Fund, and other important sectoral stakeholders, as well as the findings of the Project Steering Committee, it was decided to recommend one of the proposed models, i.e., the mixed model (Figure 2).

### 1.3. A Functional Model with the Coordinating Role of the RCOI with the Support of the National Health Fund and AOTMiT

In the model cited, it was proposed to create a new organizational unit at the level of Voivodeship Offices, in the departments responsible for health matters, the Regional Center for Investment Assessment (pol. RCOI), whose main task is to be the opinion of HB-HTA reports prepared by hospitals. In the assumed solution, individual functions were also assigned to the National Health Fund (education and promotion) and AOTMiT (standardization of methodology). This model indicates that the RCOI is to play the role of the main actor for the HB-HTA process at the regional level, where it is to integrate the concept of evaluation of health technology in one province. The selection of the Voivodeship Offices was made based on the activities of this office in the field of document analysis under the Evaluation Instrument of Investment Motions in Health (pol. IOWISZ), also because the Voivodeship Offices are not founding entities for hospitals, and hence the process of evaluating investments in healthcare units can be carried out in a transparent manner. This unit was also selected due to medical data collected by the Office in connection with the creation of Health Needs Maps, which indicate health challenges in the region. This may mean that at the Voivodeship level, it will be possible to identify planned investments in health facilities based on reliable information. At the regional level, a Voivodeship Office Health Department will play a role of arbitrator for the healthcare system and HB-HTA may be a step towards introducing such a change. This perspective convinced experts to involve Voivodeship Offices in the presented model (Figure 3).

Due to the creation of a new unit at the level of Voivodeship Offices, it is necessary to examine how RCOI can function within these units. Therefore, it was decided that the presented concept should be confronted with the activities of the Health Departments in Voivodeship Offices, which will constitute an organizational, human resources, and intellectual reservoir for the Regional Center for Investment Assessment.

The aim of the article is to evaluate the possibility of implementing a functional model with the coordinating role of RCOI with the support of the National Health Fund and AOTMiT in Poland and to indicate the strengths and weaknesses of regional coordination for HB-HTA made by Voivodeship Offices.

## 2. Material and Methods

The method used in this work is semi-structured interviews with representatives of the Departments responsible for health in Voivodeship Offices in 16 Voivodeships. All interviewees were liable for IOWISZ application analysis. The first three interviews (with representatives of the Health Departments at the Voivodeship Offices of the following three Voivodeships: Lesser Poland, Mazovian, and Lower Silesian) were conducted in June 2021 as part of the above-mentioned HB-HTA project. In order to recruit the remaining respondents for the study, the following activities were performed: (1) a letter explaining the principles of the design and model was sent to the remaining 13 Voivodeship Offices in Poland; (2) after telephone contact with a representative of the Voivodeship Offices, a multimedia presentation with a description of the project along with the most important assumptions of the model and a questionnaire in a separate file was sent to the official e-mail inboxes; (3) on the basis of a telephone conversation and e-mail correspondence, the form in which the respondents would answer the questions was selected (orally via MS Teams or in writing via MS Word). All the above-mentioned activities were carried out from October to December 2021. Interviews with representatives of four Voivodeships recorded via MS Teams lasted around 30 min each. The time taken to complete the Word file by interviewees in remaining interviews was not counted.

A major limitation of the research conducted was the inability to conduct interviews with representatives of 16 Voivodeships, due to the refusal to participate in the study by eight units. Refusals most often resulted from the heavy burden of duties of the Voivodeship Office (mainly in connection with the COVID-19 pandemic), as well as the initial analysis indicating that the HB-HTA subject matter should not be located at the level of Voivodeship Offices due to the lack of any premises for the possibility of carrying out such tasks. The lack of willingness on the part of the Voivodeship Offices also resulted from the misunderstanding of the project and the lack of dissemination of this subject among representatives of these institutions. Three Health Departments withdrew from participating in the interviews after the initial analysis of the project material. In the case of the remaining units, the waiting time for a response from the Office was too long. Previously, the very idea of the HB-HTA project was known to four Voivodeships: Lesser Poland, Lower Silesian, Mazovian, and Pomeranian. The first three of them participated in the trainings as part of the consortium’s activities

Ten interviews were conducted with representatives of eight Health Departments in the Voivodeship Offices (Health Department employees from Lesser Poland, Lower Silesian, Mazovian, Lodz, Warmia and Mazury, West Pomeranian, Pomeranian, and Lubelskie). Six of them were held in real time using the MS Teams application, and in the four remaining cases, the respondents themselves answered the questions contained in the scenario through answers saved in MS Word. Online interviews were conducted by D.M., I.K.B., and M.F. In the case of recorded calls, they were transcribed by M.F. All recordings were made by D.M, I.K.B., and M.F. and are properly deposited. The people who participated in the survey agreed to share their responses (Table 1).

The prepared topic guide included nine questions (along with sub-items extending the answers) on the possibilities of implementing HB-HTA in Poland and the role of the Voivodeship Offices in this process (the interview topic guide is available in the Appendix A to this paper).

All interviews were translated into English due to the need to analyze the material using NVivo 12. To organize the content analysis, codes were assigned to individual fragments of the interviews related to the issues arising from the questions posed during the interviews. There are seven codes related to the following issues:definition of medical innovation;HB-HTA from the perspective of Voivodeship Offices;the biggest barriers to the implementation of HB-HTA at the level of the health care system;the biggest barriers to implementing HB-HTA at the level of Voivodeship Offices;negative effects of HB-HTA implementation;the best institutions to implement HB-HTA;motivation of hospitals to implement HB-HTA.

As part of the use of the analytical program for each of the above-mentioned interviews, the so-called internal codes (using the “Autocode” function) made it possible to extract the results of the interviews even more. Thanks to this, it was possible to determine the frequency of appearance of particular phrases in the respondents’ statements as well as to indicate what information the respondents most often provided in their responses (Word-Frequency and Hierarchy Charts functions) (Table 2). Based on the assumed codes, it was possible to specify the most important information indicated by officials from Voivodeship Offices.

## 3. Results

### 3.1. Definition of Medical Innovation

Respondents identified medical innovation with the concept of novelty, implementation of diagnostic and therapeutic solutions, purchase of medical equipment, or medical patient service points. In addition, many respondents also defined medical innovation as a new organizational activity (Warmia and Mazury Voivodeship Office):

“*Medical innovation is the implementation of a new process*” (A.P.)

Respondents perceive medical innovation as a way to better organization and more access to healthcare services for patients (Mazovian Voivodeship Office). They highlighted the role of new methods of treatment and therapeutic methods as key for hospital management.

The respondents indicated that innovation is synonymous with the concept of changing the approach to an action or a process. The introduction of innovation itself aims to create the possibility of solving the problem. The purpose of implementing innovations was, among others, the increase in effectiveness (Mazovian Voivodeship Office) or improvement of the functioning of current solutions, e.g., in treatment (Lubelskie Voivodeship Office):

“*Medical innovation is any procedure, process, or organizational change in a given unit that differs from the previous method of operation*” (M.L.)

### 3.2. HB-HTA from the Perspective of Voivodeship Offices

The respondents indicated that the introduction of HB-HTA to the Voivodeship Office could significantly improve the coordination of the healthcare system at the provincial level (Lesser Poland Voivodeship Office). By using HB-HTA, Voivodeship Offices will have more information on investments in networks (Mazovian Voivodeship Office):

“*Voivodeship Offices will have more information on investments in networks. To facilitate the operation of the functions you want to run for the region, to prevent information loss at the regional level*” (U.K.)

In connection with the assumption resulting from the model, much was drawn to the implementation of HB-HTA into the IOWISZ system (pol. Instrument Oceny Inwestycji w Ochronie Zdrowia), which is currently present in Voivodeship Offices. Otherwise, HB-HTA can be introduced to the IOWISZ methodology, but it is made by the Ministry of Health, not Voivodeship Offices directly (Lodz Voivodeship Office). In addition, IOWISZ should be renewed (Lubelskie Voivodeship Office):

“*IOWISZ as an evaluation method should be more detailed. This methodology should be expanded*” (T.P.)

The respondents pointed out, however, that the methodology contained in the IOWISZ is not sufficient to properly assess the implementation of investments in hospitals. There is no way to assess the investment, at least from a clinical point of view. Voivodeship Office employees check the correctness of the application from a formal point of view or focus on economic and management aspects (Lubelskie Voivodeship Office):

“*The assessment method under IOWISZ only gives us the opportunity to check what is in the application and what is not (zero-one choice). It is not possible to evaluate the assessed information. We do not have the tools to take into account health technologies efficiency or safety*” (T.P.)

As part of the answer to the question about the prospect of HB-HTA implementation, the respondents indicated the initial barriers that currently prevent the implementation of this solution. The respondents emphasized that it is currently impossible due to the small number of people employed in the office and the low level of knowledge of employees of Voivodeship Offices on HTA, due to a different professional profile (e.g., legal education). The participation of the Voivodeship Office in the activities of HB-HTA indicates the possibility of increasing the rationality of decision-making in healthcare, while it is still an unknown topic for the officials: (Mazovian Voivodeship Office, Lesser Poland Voivodeship Office, Pomeranian Voivodeship Office):

“*As for today, Voivodeship Offices are not ready to play the role in the presented HB-HTA model*” (J.B.)

### 3.3. The Biggest Barriers to the Implementation of HB-HTA at the Healthcare System

The respondents indicated that among the biggest problems related to the implementation of HB-HTA in the Polish health care system are the lack of adequate staff at the hospital level, lack of financial resources, resistance to changes, legal problems. There is no way to implement HB-HTA without relevant courses and workshops (Warmia and Mazury Voivodeship Office, West Pomeranian Voivodeship Office, Lodz Voivodeship Office, Mazovian Voivodeship Office):

“*Lack of an appropriate legal environment for the development of an innovative approach*” (I.K.)

### 3.4. The Biggest Barriers to the Implementation of HB-HTA at the Level of Voivodeship Offices

According to the respondents, the problem in the development of HB-HTA at the Voivodeship Office will most likely be the lack of appropriate employees’ qualifications. To achieve this, a series of trainings and courses would be needed to prepare officials to fulfill their role. There is no adequate education among people working in Voivodeship Offices (Mazovian Voivodeship Office, Lubelskie Voivodeship Office, Pomeranian Voivodeship Office):

“*Lack of human and financial resources*” (M.K.)

Respondents emphasized that Voivodeship Offices are not prepared to play responsible role projected in the HB-HTA model without external assistance. Their own personal resources are not sufficient to fulfill current tasks and suffered from lack of professionals. They have noticed that Voivodeship need additional human resources from HTA institutions. Currently, they need appropriate preparation by other health stakeholders to be able to interpret data derived from hospitals. 

### 3.5. Negative Effects of Implementing HB-HTA

Among the most common side effects of the implementation of HB-HTA, respondents indicate, among others, increased competition between settings, which could become unhealthy, therefore. Moreover, additional employment is associated with the costs of maintaining staff, which may be unfavorable for both the Voivodeship Offices and the hospitals themselves. As a result of this, the costs of the investment itself will increase because of administrative activities. In addition, it was once again emphasized that there may be a shortage of people who could deal with the subject of HB-HTA. At the same time, respondents indicated the risk of a lack of success in the implementation of medical technologies. They also indicate that HB-HTA has the potential to be developed in large clinical hospitals with the appropriate potential and qualifications (Mazovian Voivodeship Office, Lodz Voivodeship Office, Mazovian Voivodeship Office):

“*Clinical hospitals will be interested in implementing HB-HTA. There may be an even greater difference in access to modern technologies between clinical hospitals and district medical entities, with less ability to introduce innovation*” (I.K.)

### 3.6. The Best Institutions for HB-HTA Implementation

Respondents most often indicated that the institutions that should implement HB-HTA are the National Health Fund, the Agency for Health Technology Assessment and Tariffs, and the Ministry of Health. All of these units were considered the most competent in the field of HTA due to their human resources and the information collected in the databases. Furthermore, due to the involvement of the National Health Fund, it will be possible to introduce financial incentives related to the implementation of HB-HTA. The Ministry of Health was recognized as an important player due to its role as a legal regulator of the healthcare system. In this way, the respondents also indicated the benefits of an in-depth analysis of investments made in hospitals. The respondents did not mention institutions such as universities or research centers, for example. In most cases, the main roles of the above-mentioned institutions should include coordinating the entire process in the territory of the country and performing the function of the appropriate project leader due to the competences in these institutions. In addition, these institutions should have financial resources and experience, hence the respondents listed the National Health Fund, AOTMiT, or the Ministry of Health (Warmia and Mazury Voivodeship Office, Mazovian Voivodeship Office, Lower Silesia Voivodeship Office):

“*The National Health Fund can introduce a benefit on the list of guaranteed benefits, which will be assessed by HB-HTA*” (A.D.)

“*AOTMiT should have the greatest impact on the implementation of this solution*” (M.L.)

### 3.7. Motivation of Hospitals to Implement HB-HTA

Thanks to HB-HTA, the competitiveness of institutions can increase. Respondents also indicated that hospitals can obtain new certificates. In addition, it is imperative that hospitals, together with HB-HTA, obtain additional funds for their operation from the National Health Fund. Respondents indicated that it is necessary to introduce additional points when evaluating a given facility in the certification process resulting from the HB-HTA. At this point, it was also noted how important it should be to create a number of potential benefits, mainly in the form of financial bonuses, when encouraging outlets. All the participants in the talks pointed to financial benefits. The very idea of HB-HTA, according to the participants, should result from the activity of the hospital management representatives. They must be leaders in HB-HTA introduction (Mazovian Voivodeship Office, Lubelskie Voivodeship Office, Lodz Voivodeship Office)*:*

“*I think that financial matters should definitely be a motivator and should be taken into account when financing from the National Health Fund.*” (U.K.)

“*A wide-ranging information campaign showing the potential benefits of implementing HB-HTA should be sent to hospitals; creation of trainings in this field for managerial staff or other people who could support the development of HB-HTA in the units may also be beneficial*” (I.K.)

## 4. Discussion

Based on interviews with representatives of the Health Department in the Voivodeship Offices from half (8) of the Polish Voivodeships, it can be indicated that they are not sufficiently prepared to perform their functions in accordance with the tasks set by the authors of the model. The model presented in the project cannot be implemented due to lack of human resources and experience in the HTA field among workers of analyzed institutions. Voivodeship Offices are underfunded, and they lack funds to hire new, highly qualified workers with relevant knowledge. The presented barriers and limitations make it clear that the target model should be based to a greater extent on the activities of central institutions, which, in the respondents’ opinion, are characterized by higher competences to perform supervisory functions and have the necessary human resources. It is believed that national institutions, such as the National Health Fund or the Agency for Health Technology Assessment and Tariffs, are better able to act as institutions implementing HB-HTA in Poland. Representatives did not mention universities in the model. This could have resulted from being more familiar with institutions as national payer and Ministry of Health. It is possible that respondents considered the governmental agenda because they are derived from public agenda institution themselves and they lack deeper contact with academic environment. In terms of public health tasks, Polish universities are not deeply engaged by public institutions. Academics in Poland play a role of consultants to specific tasks, but healthcare entities have not appointed them to make systematic reviews or economic evaluations in terms of investments. 

It should be noted that a centralistic approach to Hospital-Based Health Technology Assessment is rare among countries that have implemented HB-HTA in their practice. Mostly, models in which regional or local institutions are responsible for evaluating reports from hospitals are used. Furthermore, scientific organizations such as universities and medical associations (associating physicians and other healthcare professionals) show particular interest in the methodology of evaluating investments and undertakings in hospitals [16]. The functioning of HB-HTA units as part of cooperation networks in many countries is based on the methodology prepared by doctors, academics, and healthcare managers. Examples of the existence of Hospital Health Technology Assessment are often characterized by the operation of various entities in a specific administrative area (province or region). This is because HB-HTA, defined essentially as small HTA (mini HTA), aims to quickly present the conclusions of the conducted analyses. These studies are often based on short systematic reviews and are unsystematic because the most important thing is to make rational managerial decisions in a short time [17,18,19]. Adopting HB-HTA helps to stabilize hospital budgets and provides increasing influx of new technologies to make prioritization a necessity and may lead to improved patient safety [20].

The HTAi Hospital Based Health Technology Assessment sub-interest Group has proposed a framework for classifying the various types of HTA activities that exist at the healthcare organization level, depending on institutional and other socio-economic factors characterizing the healthcare systems in different countries. From these variables, four different approaches to HB-HTA can be identified: the ambassador model; the mini-HTA; the internal committee; and the HTA unit. Each approach has specific purposes, structures, strengths, and weaknesses [19]. They differ from level of formalization, entities engaged in the model, and type of healthcare professionals working within these groups. The rich variety of HB-HTA models means that there is no single template solution for hospitals. Because of that, HB-HTA is often an internal hospital employees initiative limited to few healthcare entities. There is no one pattern that could be implemented in all healthcare systems. Otherwise, it is worth mentioning that among the countries where hospital health technology evaluation is active, Spain is particularly engaged. The elaboration of HB-HTA reports in hospital units is then scrutinized by the authorities of some Spanish provinces. Only a few countries prepared HB-HTA with representatives of national and local authorities [21].

Most countries that have already implemented the HB-HTA models have a decentralized approach to managing the health sector. Due to this structure, it was easier to implement a solution in which the decision-making level is carried out by lower administrative units. It was indicated that the management of the implementation of medical technologies was delegated to such a level that it is easier to contact individual decision-making levels [21].

Because HB-HTA has developed to the greatest extent in the countries of western Europe and North America, it should be considered whether such a solution on a similar scale can be implemented in Poland. Based on that publication and the information that has been collected and analyzed as part of the project, one can get the impression that in Poland institutions at the regional level (both central and local) are not too eager to give opinions on investment applications in accordance with the HB-HTA methodology [22]. From the international perspective it is worth mentioning that there is still not enough knowledge and number of trainings among European healthcare professionals in terms of HTA is not sufficient. Future skills implementation programs will need to pay particular attention to the training needs of all healthcare related workers involved in the use of health technologies and in their assessment process [23]. Employees must be trained firstly in order to analyze a hospital’s investment reports in accordance with HB-HTA methodology. 

The possibilities of the Polish Voivodeship Offices involvement in the process of giving opinions on the HB-HTA report produced in the hospital should be considered low. They should be equipped with knowledge and skills during long HTA workshops by HTA professionals. Without adequate background, assessment performed by Voivodeship Offices workers could be incorrect and affect the investment activities in hospitals.

Moreover, the project idea itself was born at the central level, among the decision-makers of the Ministry of Health and the National Health Fund, scientists from the Lazarski University, and the National Institute of Cardiology. Hospital centers from the largest Polish cities, such as Warsaw, Cracow, Gdansk, and Wroclaw, played an active role in the implementation of the project. District hospitals were not represented in the implementation of design phase of the HB-HTA in Poland. Appropriate know-how is implemented at the level of large clinical hospitals in provincial cities. 

This work has several limitations. The most important is the lack of response from all Voivodeship Offices in Poland. It caused the lack of inferring conclusions to all Voivodeships in terms of HB-HTA coordination. Furthermore, the set of qualitative questions allowed for a free approach to the answers, making it impossible to fully standardize the topic guide. The study was also carried out during the COVID-19 pandemic, which made it impossible to hold longer talks due to the high workload of employees of Voivodeship Offices. The interviews were conducted at different time intervals by different researchers, which may have contributed to reducing the consistency of the results obtained. Additionally, Voivodeships are different in terms of scale, number of hospitals, resources in Health Departments, and troubles with local healthcare entities debts. All of them implement IOWISZ based on the same methodology in all Voivodeships, but other factors could be various and have impact on health technologies evaluation. Moreover, it would be worth using Delphi methodology and organizing an expert panel (8–12 people) with healthcare professionals. It would strengthen making conclusions to complete knowledge regarding Hospital-Based Health Technology Assessment. That methodology should be involved in the further analysis to get more detailed data.

The strengths of the presented study include the fact that Voivodeship Offices carry out tasks related to the area of health through their activities (e.g., IOWISZ), therefore they are already partially involved in the process of monitoring hospital investments in the voivodeship. The implementation of HB-HTA would be aimed at deepening the application of the methodology and strengthening the cooperation between hospitals and Voivodeship Offices. Influence on HB-HTA from the regional authority might be reflected in the delivery of information for each HB-HTA report submitted by hospital for the opinion of Voivodeship Offices about the competitive potential of a particular technology in the region, to eliminate the relatively frequent duplication of investments close to each other. If the barriers indicated by the employees of these units were to be removed or reduced, the Voivodeship Offices could, in the proposed HB-HTA model, act as a coordinator of investment activities of medical institutions in each Voivodeship. Hence, the following analysis, by indicating these barriers, may become the beginning of the implementation of reforms in the functioning of Health Departments of Voivodeship Offices, so that the coordination task can be carried out by these entities. There are possibilities within these structures for the introduction of HB-HTA, but now there are too many limitations; consequently, the proposed model for the introduction of HB-HTA is not recommended at the moment. 

## 5. Conclusions

Based on the results of the study, it can be concluded that there are no units that should play a supervisory role in the process of evaluating HB-HTA reports from hospitals. These institutions lack the appropriate human and financial resources to create an appropriate unit, the Regional Investment Assessment Center. The study group was limited to half of all Voivodeship, and that factor prevented holistic results generalization. However, the rest of Voivodeship Offices refused taking part in the study due to lack of time and interest/knowledge about HB-HTA. It can be presumed that they were not interested in that topic or were not familiar with the HB-HTA concept. It should also be considered that our study was conducted during the pandemic, and the Voivodeship Offices had important crisis management responsibilities in the region, which naturally required focusing on these aspects.

The participation of the Voivodeship Office in the HB-HTA process is not ruled out after the formal extension of the tasks of this level of state administration to activities related to HB-HTA described in the strategic functional model and the allocation of additional financial resources to these tasks. The employees of the Voivodeship Office made the reservation that it is impossible to perform the current duties with a very low level of staff, and new people with appropriate qualifications should be employed to perform the tasks described in the model. Hence, HB-HTA can be implemented with significant funding from the Health Departments of the Voivodeship Offices. As has been shown above, once the barriers are eliminated, the Voivodeship Offices can provide opinions on HB-HTA reports indicating their methodological correctness/incorrectness and the competitive potential of a given technology.

The decision to entrust the role of the HB-HTA process coordinator at the Voivodeship level to the Voivodeship Office is an administrative decision and should be taken at the central level with the participation of the Minister of Health.

## Figures and Tables

**Figure 1 ijerph-19-11235-f001:**
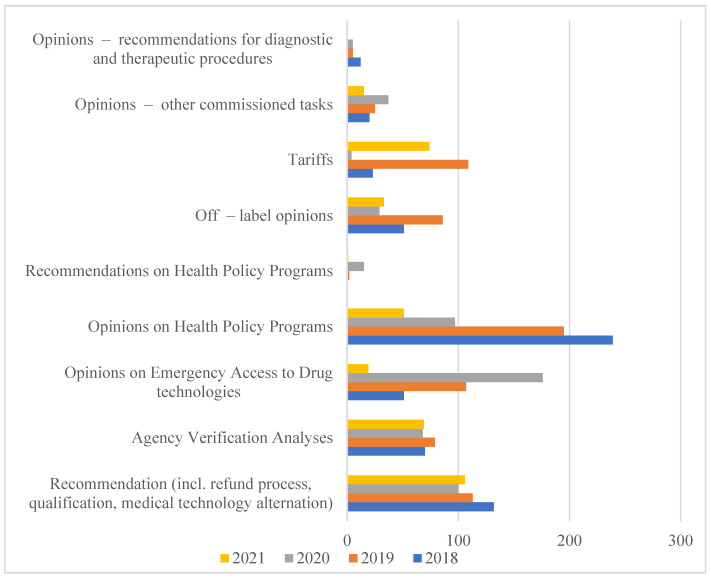
The Polish HTA Agency orders performer in years 2018-2021. Source: AOTMiT Statistics in years 2018–2021. Warsaw, July 2021. https://www.aotm.gov.pl/o-nas/statystyki-aotmit/. Access date: 18 June 2022.

**Figure 2 ijerph-19-11235-f002:**
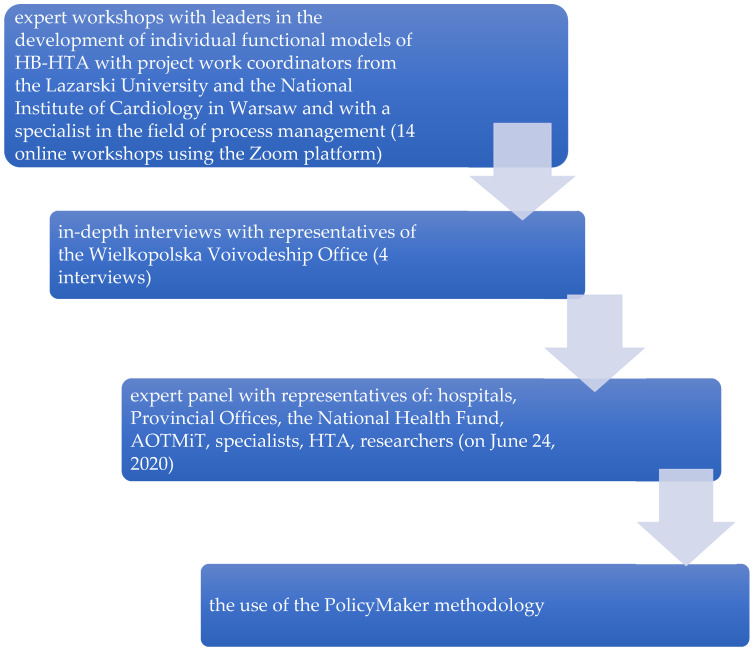
Methods used to create “mixed” model. PolicyMaker (PM), produced by Michael R. Reich is based on rich theoretical foundations, where the author mentioned: the model of rational behavior, the model of public administration, bureaucratic model, concepts of interest groups, model of public choice. Political mapping is primarily used for identification entities (stakeholders, shareholders, actors) that may have an impact on the implementation of a complex program (changes, reforms), recognizing the relationships that occur between them with regard to the opportunities for implementation of a given change, reform. Source: Kowalska-Bobko I. Gałązka-Sobotka M. HB-HTA functional model with the coordinating role of the Regional Center for Investment Assessment supported by the National Health Fund and the Agency for Health Technology Assessment and Tariff System mixed model. “Public Health and Management” 2020; 18 (4). P. 298–309.

**Figure 3 ijerph-19-11235-f003:**
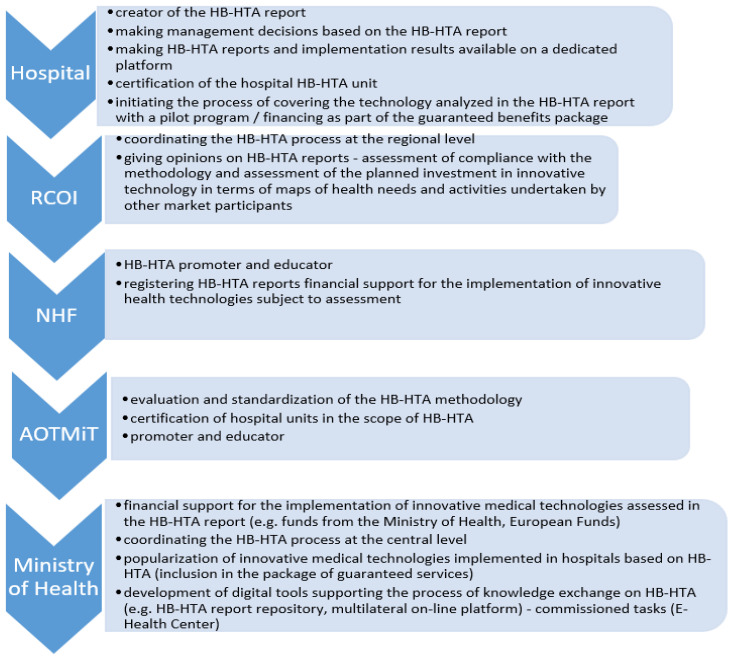
Roles in the HB-HTA process assigned to key entities in the functional strategic model with the coordinating role of the RCOI supported by the National Health Fund and AOTMiT (mixed model). Source: Kowalska-Bobko I. Gałązka-Sobotka M. HB-HTA functional model with the coordinating role of the Regional Center for Investment Assessment supported by the National Health Fund and the Agency for Health Technology Assessment and Tariff System mixed model. “Public Health and Management” 2020; 18 (4). P. 298–309.

**Table 1 ijerph-19-11235-t001:** Study characteristics.

Name of Voivodeship	Acceptance/Refusal/Lack of Response	Period of Interview	Form of Interview	Interviewees Initials
Lower Silesian	Acceptance	June 2021	MS Teams	I.D.S. and A.B.
Kujawsko-Pomorskie	Lack of response	–	–	–
Lubelskie	Acceptance	October 2021	MS Teams	T.P.
Lubuskie	Lack of response	–	–	–
Lodz	Acceptance	November 2021	MS Word	I.K.
Lesser Poland	Acceptance	June 2021	MS Teams	J.B.
Mazovian	Acceptance	June 2021	MS Teams	U.K. and A.P.
Opolskie	Lack of response	–	–	–
Podkarpackie	Refusal	–	–	–
Podlaskie	Refusal	–	–	–
Pomeranian	Acceptance	November 2021	MS Word	M.K.
Silesian	Lack of response	–	–	–
Świętokrzyskie	Refusal	–	–	–
Warmia and Mazury	Acceptance	December 2021	MS Word	M.L.
Wielkopolskie	Lack of response	–	–	–
West Pomeranian	Acceptance	November 2021	MS Word	K.G.M.

**Table 2 ijerph-19-11235-t002:** The most frequently repeated words in all analyzed codes.

**Definition of Medical Innovation**		
**Word**	**Count**	**Weighted Percentage (%) ***
medical	14	7.04
new	14	7.04
innovation	8	4.02
Given	6	3.02
process	6	3.02
**HB-HTA from the perspective of Voivodeship Offices**		
**Word**	**Count**	**Weighted Percentage (%)**
HTA	126	3.81
health	114	3.45
implementation	66	2.00
assessment	63	1.91
role	51	1.54
**The biggest barriers to the implementation of HB-HTA at the healthcare system**		
**Word**	**Count**	**Weighted Percentage (%)**
health	23	3.87
medical	21	3.54
funds	17	2.86
financing	14	2.36
probably	13	2.19
**The biggest barriers to the implementation of HB-HTA at the level of Voivodeship Offices**		
**Word**	**Count**	**Weighted Percentage (%)**
education	35	2.35
office	35	2.35
staff	31	2.08
knowledge	28	1.88
health	23	1.54
**Negative effects of implementing HB-HTA**		
**Word**	**Count**	**Weighted Percentage (%)**
health	25	3.68
HTA	19	2.79
staff	15	2.21
assessment	14	2.06
technology	13	1.91
**The best institutions for HB-HTA implementation**		
**Word**	**Count**	**Weighted Percentage (%)**
health	25	3.68
HTA	19	2.79
staff	15	2.21
assessment	14	2.06
technology	13	1.91
**Motivation of hospitals to implement HB-HTA**		
**Word**	**Count**	**Weighted Percentage (%)**
hospitals	8	2.65
additional	7	2.32
financial	6	1.99
HTA	6	1.99
opinion	6	1.99

* Weighted percentage is the frequency of the word relative to the total words counted in the code.

## Data Availability

The data presented in this study are available in the article and Appendix A. More data will be made available on a reasonable request by contacting the corresponding author.

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
