# Peer review of "Possibilities of Implementing Hospital-Based Health Technology Assessment (HB-HTA) at the Level of Voivodeship Offices in Poland"

_ijerph, 2022, doi:10.3390/ijerph191811235_

Round 1

Reviewer 1 Report

Generally, I think this article is interesting in that it describes the situation of HTA in Poland, particularly HB HTA but it needs to attend to the needs of the international readership, with more appropriate vocabulary at various points in the article. Some references are also in Polish, so it would help if they were translated.

I think it is important to describe the healthcare context in Poland – how is healthcare financed? Is it entirely from the public purse or a mixed model of public and private? How is healthcare funding managed – it sounds like it is mainly or all from central government.

I suggest that this work could be converted to a commentary or opinion piece rather than a research article, because the research aspects are very thin, e.g., there is very little description of the sample characteristics, ethical review, whether/how the various interviewers collaborated in the analysis.

Some of the citations are in superscript while others are not.

Abstract: The Health Technology. The 2nd sentence should state that Poland is the site of the study.

There is also a notable lack of understanding of the difference between qualitative and quantitative research methodologies. Using the term ‘questionnaire’ implies a survey using quantitative methods of analysis. The term ‘topic guide’ or ‘aide memoire’ might be more useful. Content analysis, which is being attempted here, involves the counting of words and phrases, but with such a small sample, it doesn’t seem applicable, and its usefulness hasn’t been demonstrated.

‘The material was divided into seven codes’ – this indicates a lack of understanding of the process of qualitative analysis. Either that, or the translation from Polish is lacking. In any case, the method of analysis needs to be stated, i.e., qualitative content analysis, or thematic analysis, etc., with references. Codes are generated from the analysis of the texts.

Introduction

Line 37 - ‘a state organizational unit with legal personality functions’

Line 42 – analyses

Chart 1 – Opinions ‘on’ not ‘to’; performers? Performance

Line 53 – ‘in this area’ – vague – area of work or regional area?

Line 65 – allows to present the thesis? – Unclear meaning

Line 89 – territorially competent communes and counties – not sure what this means

Line 102 ff – it would help the reader to have a flow diagram to depict the progression of the HB-HTA, from the creation of pilot units and networks, expert meetings, and the resultant functional ‘mixed’ model.

Line 127 –When you say ‘opinion’, do you mean only commenting or also making policy recommendations based on the results of the HTA report from the hospitals?

Line 130 ff – I see the value and logic of this 'mixed' model, integrating the regional and national levels, but I wish there was more detail on it.

Line 141 – I think you mean that the ‘structure’ will play the role of an ‘arbitrator’ or ‘arbiter’

Figure 1 – it would help the reader to know if these roles were carried out at the regional/local or national/central level etc. There are a lot of acronyms, and it can get confusing. A glossary would also help. ‘The author’s own study’ – what study is that? Any reference? If it is not published, it is not necessary to state that it is your own study – because which author is it?

Line 151 – unclear

Line 154 – the aim of the article should be to present the results of the evaluation that has been undertaken – it’s not clear what kind of evaluation it is, although the aims are stated. On further reading, it sounds like an exploratory study, or even a consultation exercise.

Line 159 – As mentioned, more detail needed on the method. If they are in-depth interviews, could you give us an idea of how long interviews lasted? Who are these ‘representatives’ – and how were the 1st three representatives recruited? Convenience sampling? Purposive sampling?

Line 169 – are you saying that some filled in a form answering your questions? If so, how can this be classed as ‘in-depth’? (Sorry I can’t open your supplementary folders – you should be able to provide an excel table of your codes from NVIVO – or present it in Word).

Line 175 – Refusesals

A table describing the participants/interviewees, how they were recruited, and decliners, and reasons for declining would be useful for the reader. If possible, their voivodeships could be included in the table if this is not too sensitive.

Line 185ff – those who had training (not trainings) – could also be included in the table. Which interviews were online, and which were recorded and transcribed, as well as the interviewers’ initials could be included?

Line 196 – I am confused – was there a survey as well as an interview?

Line 201 – one doesn’t ‘organise the content analysis’ – the analysis enables you to organize and systematize the data.

Were the different interviewers involved in the qualitative analysis at all?

Line 217 – what is meant by ‘assumed codes’?

Line 239 – ‘much was drawn’ – do you mean that there were a lot of references to that topic?

Line 239ff – you seem to be describing a clash of two systems, but IOWISZ was not referred to earlier in the introduction and I am not sure what it is.

Paragraphs beginning Line 246, Line 251, and Line 264 - When you refer to ‘the respondents’ - how many said these things? Could you say they were the majority?

Line 282 – these ‘side effects’ (outcomes?) are hypothetical – but you could talk about likelihood, or possible risks or negative impacts described by your interviewees

Line 295 – these are recommendations made by your respondents

Line 305 – why were universities etc. not mentioned, do you think? This question needs addressing in the Discussion.

Line 317 – also recommendations. Financial rewards rather than ‘benefits’?

Line 360-361 – ‘It was indicated’ – by whom? ‘individual decision-making levels’ – decision-makers? Policy makers?

Line 364 – publication ‘below’? where?

Line 374 – is the term ‘Lower level hospitals’ appropriate? Do you have such terms as tertiary, secondary, or primary care, or general hospitals or district hospitals – so that the international readership can relate better to what you mean? Or acute care, or specialty or teaching hospitals?

Line 379 – One doesn’t ‘standardize the questionnaire’ – the same questions or topics are applied in all the interviews, but the answers hopefully will reflect a range of responses through their ‘open’ answers – which makes for interesting comparisons. These comparisons have not been forthcoming in your analysis. For e.g., were there different responses, and can you account for these differences? I am guessing that voivodeships differ in size and styles of management depending on their constituencies.

Line 385-6 – based on 10 interviews out of 8 voivodeships of which there are 16 altogether, you surely cannot generalize. What if the other 8 gave conflicting responses if you managed to interview them? You can suggest probabilities and likelihoods, however. In your conclusion, are there any lessons for the wider readership and for policy makers to take away from your ‘study’?

Author Response

Dear Reviewer,

firslty, I would like to thank you for your assistance in terms of article. Please find attached Word file with my notes to your commentaries. I have respected to all your comments in that file.

Best regards,

Author

Reviewer 2 Report

The article is very interesting and focuses on a current issue of great interest for Public Health and health systems.

The authors focused on a fundamental and current issue linked to the implementation of the introduction of technological innovation in the hospital setting and the related evaluation process.

Health technology assessment (HTA) is increasingly performed at the local or hospital level where the costs, impacts, and benefits of health technologies can be directly assessed. Although local/hospital-based HTA has been applied in recent years in several countries, barriers to its implementation still persist.

Therefore, the aim of this article is to evaluate the possibility of implementing a functional model in Poland and to indicate the strengths and weaknesses of regional coordination for HB-HTA.

This study could give an important scientific contribution on these issues, however it has several limitations.

Therefore, major revisions are suggested. In particular:

-          The introduction should be integrated by further emphasizing the importance of HTA as a value-based and evidence-based governance tool to support decision-making in healthcare (Cyr PR, et al. Evaluations of public health interventions produced by health technology assessment agencies: A mapping review and analysis by type and evidence content. Health Policy. 2021 Aug;125(8):1054-1064. doi: 10.1016/j.healthpol.2021.05.009.; Henshall C, Schuller T; HTAi Policy Forum. Health technology assessment, value-based decision making, and innovation. Int J Technol Assess Health Care. 2013 Oct;29(4):353-9. doi: 10.1017/S0266462313000378).

The Authors should also give more information on HB-HTA (read for example: Gagnon MP, Desmartis M, Poder T, Witteman W. Effects and repercussions of local/hospital-based health technology assessment (HTA): a systematic review. Syst Rev. 2014 Oct 28; 3:129. doi: 10.1186/2046-4053-3-129).

-          The methodology should be strengthened. The Authors have stated the objective limits and difficulties of their study, however I would suggest completing their work by applying the Delphi methodology. Delphi is a method used to reach, though an iterative process, a consensus on a specific topic among experts with knowledge and experience in that area (Niederberger, M.; Spranger, J. Delphi technique in health sciences: A map. Front. Public Health 2020, 8, 457). The minimum acceptable number to ensure the validity of a Delphi process is between 8 and 12 experts (Hallowel, M.R.; Gambatese, J.A. Qualitative research: Application of the Delphi method to CEM research. J. Constr. Eng. Manag. 2010, 136, 1–9).

-          The results show the main inputs received by the interviewees but in a general way. Perhaps it might be useful to insert some percentage at least for the common answers.

-          In the discussion, the Authors could comment on some results such as the barriers indicated for the implementation of HB-HTA in health systems or at the level of Voivodeship Offices. For example, training in HTA is a current problem that requires effective interventions for the implementation of this multidisciplinary approach (Hoxhaj I et al. HTA Training for Healthcare Professionals: International Overview of Initiatives Provided by HTA Agencies and Organizations. Front Public Health. 2022 Feb 10; 10:795763. doi: 10.3389/fpubh.2022.795763). Again the barriers related to financial resources and resistance to changes could be further argued. The Authors could also propose improvement actions and interventions to overcome the criticalities that emerged from their interviews.

-          Eventually, The Authors' paper is really very interesting and highlights several barriers that hinder the application and implementation of HB-HTA. In an era of a growing economic pressure for all health systems, evidence based approaches such as HTA are needed both to invest and to disinvest in health technologies (Calabrò GE, La Torre G, de Waure C, et al. Disinvestment in healthcare: an overview of HTA agencies and organizations activities at European level. BMC Health Serv Res. 2018 Mar 1;18(1):148. doi: 10.1186/s12913-018-2941-0). Therefore, studies such as the one proposed deserve to be shared with the scientific community also to promote the application of HB-HTA.

Author Response

Dear Reviever 2,

I am kindly thankful that you spend your time to analyse my research. With regard to your commentaries by lengthening introduction and adding new paragraph I figured a new paragraph in. It started by phrase: With the growing use and importance of health technology assessment (HTA) in decision making during recent years, health technology assessors, decision makers and stakeholders are confronted with methodological challenges due to specific characteristics of health technologies ...". I highlited the role of HTA in policy-making process and EBM adoption.

Unfortunately, I cannot add the Delphi methodology to my study because of a lack of time. I promise I will use that method in my further articles. I wrote in discussion that it will be a good scientific approach to analyze data thanks to Delphi method. 

I used your suggested bibliography to add something interesting to discussion. I wrote e.g. paragraph with more deeply analysis of HB-HTA network. I quoted study regarding to HTA proficiency among healthcare workers. I added in results percentages of the most frequent phrases as you recommend.

Once again, I am wholeheartedly grateful for your guides and valuable tips.

Best regards,

Maciej Furman

Reviewer 3 Report

Small survey sample, half of the provinces refused - therefore, it is difficult to infer the entire population of local governments from the responses obtained.

Barriers to HTA implementation described perfunctorily, in too little details.

A very small part of the manuscript consists of the results.

In discussion lacked references to specific European countries and the world.

Poor bibliography.

Author Response

Dear Reviewer 3,

I would like to give an acknowledge for your comments to my study.

I added quotes to 'results' part and immerse into interviewees responses. I added the reasons of barriers to HB-HTA implementation, eg. in terms of insitutional weakness presented by Voivodeship Offices. I presented holistic perspective in terms of HB-HTA approach.I added new references to bibliography. I changed study limitations too.

Once again, thank you for comments and analysing my study.

Best regards,

Author

Round 2

Reviewer 2 Report

The authors made some suggested additions, declaring the impossibility of implementing the recommended methodology (delphi process). However, they include this aspect in the limitations of the study.

However, I would suggest making further changes, reported below:

- Review the appropriate use of acronyms throughout the paper.

 - In the introduction:

-          The authors report the HTA definition. However, they report that "Health Technology Assessment (HTA) ... consists in the evaluation of the features and effects of medical technologies, based on a systematic review ...". HTA is not carried out with a systematic review and the methodology is very rigorous. I suggest replacing with the sentence reported in the cited paper (ref 1): "HTA is a multidisciplinary process that uses systematic and transparent methods to consider the best available evidence to determine the value of a health technology at different points in its lifecycle. It consists in the evaluation of the features and effects of a health technology compared with existing alternatives, to inform decision making in order to promote an equitable, efficient, and high-quality health system".

-          Line 35: you can only report the acronym "HTA"

-          The authors supplemented the introduction with the following sentence “With the growing use and importance of health technology assessment (HTA) in decision making during recent years, health technology assessors, decision makers and stakeholders are confronted with methodological challenges due to specific characteristics of health technologies (e. g. pharmaceuticals, diagnostic tests, screening programs), their developmental environment, and their regulation process [2]. Several countries have therefore developed policy frameworks to allow access to the technology on the condition that additional evidence is generated through: “access with evidence generation” mechanisms [3]”.

The concepts reported are important but in my previous report I suggested other integrations such as “The introduction should be integrated by further emphasizing the importance of HTA as a value-based and evidence-based governance tool to support decision-making in healthcare (Cyr PR, et al. Evaluations of public health interventions produced by health technology assessment agencies: A mapping review and analysis by type and evidence content. Health Policy. 2021 Aug;125(8):1054-1064. doi: 10.1016/j.healthpol.2021.05.009.; Henshall C, Schuller T; HTAi Policy Forum. Health technology assessment, value-based decision making, and innovation. Int J Technol Assess Health Care. 2013 Oct;29(4):353-9. doi: 10.1017/S0266462313000378).

The authors should also give more information on HB-HTA (read for example: Gagnon MP, Desmartis M, Poder T, Witteman W. Effects and repercussions of local/hospital-based health technology assessment (HTA): a systematic review. Syst Rev. 2014 Oct 28; 3:129. doi: 10.1186/2046-4053-3-129)”. In the paper there is a paragraph on HB-HTA but since this is the focus of the work it is important to include some sentences also in the introduction.

-          Figure 1 is not legible and there are overlapping boxes.

In the “Material and Methods” section:

-          I suggested integrating the methodology with the delphi process. However, the authors expressed the impossibility of applying it at this stage and they include this limitations in the discussion.

-          However, the authors supplemented this paragraph with other specifications that have improved this section.

-          I suggest inserting in the text the reference to table 1.

In the “Results” section:

-          The authors supplemented this paragraph with additional data and improved this section.

-          I suggest correcting the number of tables. The table in the results section should be number 2 (line 425). Insert reference to this table in the text.

In the Discussion:

-          I would begin the discussion by taking up the objective of the study and then commenting on its main results.

-          The authors supplemented this section with additional data and improved the discussion.

-          In the part relating to the limits of the study and in particular the reference to the delphi process (“Besides….more detailed data”), I suggest to modify it as follows: “In future studies, the application of more rigorous methodologies and multi-stakeholder consultation, for example through the Delphi process, could strengthen our results and contribute to the definition of strategies to promote and implement the application of HB-HTA”.

-          After the limitations, I suggest inserting the strengths of the study according to the point of view of the authors.

In the conclusions:

-          I suggest writing the conclusions more clearly and explicitly. This last section of the paper should make us quickly understand the main results of the study, the possible implications for the Polish health system and the possible improvement actions proposed for the implementation of HB-HTA in Poland (and internationally, if possible).

-          I would write "Based on the results of this study…” and not “Based on ten semi-structured interviews with representatives of eight Voivodeship Offices…”

-          Line 540-543: The inserted sentence should be supported by evidence or adequate references because in the paper does not refer, for example, to discussions with other experts.

Author Response

Dear Reviewer,

I kindly than you for your comments. I incorporated your changes into the article. I added two references which you highlited in your review. I extended three parts: introduction, discussion and conclusions. We wrote strengths of our study and conclusions for Polish policy makers. 

I am wholeheartedly grateful for your time and commitment. 

Best regards,

Author